# Digital Technologies, Circular Economy Practices and Environmental Policies in the Era of COVID-19

**Syed Abdul Rehman Khan** [1,2]**, Pablo Ponce** [3] **, George Thomas** [4] **, Zhang Yu** [5,6]**, Mohammad Saad Al-Ahmadi** [7] **and Muhammad Tanveer** [8,*]

1 School of Management and Engineering, Xuzhou University of Technology, Xuzhou 221018, China; khan_sar@xzit.edu.cn
2 Beijing Key Laboratory of Urban Spatial Information Engineering, Beijing 100084, China
3 Carrera de Economía and Centro de Investigaciones Sociales y Económicas, Loja 110150, Ecuador; Pablo.ponce@unl.edu.ec
4 Department of Marketing, College of Business Administration, Prince Sultan University, Rafah Street, Riyadh 11586, Saudi Arabia; gthomas@psu.edu.sa
5 School of Economics and Management, Chang'an University, Xi'an 710054, China; 2020023001@chd.edu.cn
6 Department of Business Administration, ILMA University, Karachi 75190, Pakistan
7 Information Systems & Operations Management Department, KFUPM Business School, King Fahd University of Petroleum and Minerals, Dhahran 31261, Saudi Arabia; alahmadi@kfupm.edu.sa
8 Prince Sultan University, Rafah Street, Riyadh 11586, Saudi Arabia
* Correspondence: mtanveer@psu.edu.sa

**Abstract:** The degradation of the environment is associated with economic activity, particularly with the linear way in which the economy does not make efficient use of resources. However, the circular economy is opposed to this linear paradigm, since it makes the most of the resources in trying to achieve zero waste. In this context, this study investigates the relationship between industry 4.0 technologies, COVID-19 outbreak, environmental regulation policies and circular economy practices. A questionnaire is designed to collect information from 214 big and private manufacturing firms in Ecuador, and subsequently, through CB-SEM, the information is processed, and the study paths are validated. The results suggest that industry 4.0 technologies and environmental regulation policies are driving circular economy practices during the pandemic. The study finds no evidence favoring COVID-19 being a determining factor in the adoption of the circular economy. The results provide a policy framework for the adoption of a circular economy.

**Keywords:** industry 4.0; circular economy; environmental regulations; manufacturing supply chains; COVID-19



## 1. Introduction

The outbreak of COVID-19 has been the cause of the current economic crisis, stemming from the global supply chain's decline [1]. In this regard, the pandemic has shown the worst deficiencies of firms and consumers and their vulnerability to risk situations [2]. Despite this, not everything has been discouraging, since it has become clear that environmental sustainability is the way to face a situation of risk and uncertainty, which the supply chain faces [3]; however, the economy faces an economic system of linear production, in which a real challenge to achieve environmental sustainability is presented, since resources are not used efficiently, and too much waste is generated—that is, it is a model that is based on taking that makes waste [4]. On the contrary, circular economy (CE) practices focus on the efficient use of resources based on the 10Rs (reject, rethink, reduce, reuse, repair, recondition, remanufacture, reclaim, recycle and recover) [5]. The decision to adopt cleaner production systems in which the CE is accentuated has taken on a greater force since COVID-19 [6]. Consequently, the CE captures locals' and strangers' attention, since it is crucial for the future to achieve a cleaner and more environmentally-friendly production,

based on a production system with zero waste [7]. Therefore, reaching a CE is an arduous process in which economic agents' mentality must be changed to design ecological products in which cleaner production methods are used through the sustainable management of the supply chain from front to back [8]. It should be noted that entrepreneurs are agents of change that adapt to circumstances and can contribute eloquently in the transition towards a sustainable CE [9,10].

Although the CE process is very understandable in theory, the CE establishes a radical change from a linear economic model to a circular one, in which each phase of production represents a systemic change that is very complex to put into practice [11]. Generally, the implementation CE is difficult due to barriers such as lack of resources, ignorance of the innovation of the process, fear of failure, and return on investment, among others [12,13]. Otherwise, the adoption of emerging technologies from industry 4.0 (I4.0) are poised to be determinants to overcome these barriers [14,15] and to implement CE during the COVID-19 pandemic [2]. Some of the I4.0 technologies are artificial intelligence and big data, which allow firms to make better decisions when making sustainable decisions around the choice of resources [16,17].

Therefore, digital transformation is increasingly used in manufacturing industries, allowing stronger, more resilient, and intelligent production processes, which allow firms to apply CE [16]. Conversely, institutional regulation is a factor that determines the adoption of CE in the supply chain, which is why this research constitutes an element of analysis [18].

Even though emerging technologies are varied and used to achieve environmental sustainability, in developing countries, there is little evidence on the role of I4.0 in the adoption of CE during the pandemic [5]. Therefore, this document is one of the pioneers to be developed in Ecuador. In this country, the impulse for the adoption of a CE is recent; at the end of 2019, the government authorities signed the "Pact for the Circular Economy," which seeks to promote the industrialization of waste, use of renewable energy, sustained use of resources among others [19]. However, according to the careful review of the literature carried out, there are no formal investigations that examine the determinants of CE in the country, much less during the COVID-19 pandemic.

In this sense, this research aims to examine the relationship between I4.0, COVID-19, institutional regulation, and CE in Ecuador during the pandemic. Information is collected from various firms through a questionnaire which contains several questions representing each examined construct, with a response option according to the Likert Scale. Subsequently, covariance-based SEM (CB-SEM) is used to process the information collected in 214 Ecuadorian manufacturing firms that are big and private. The research findings reveal unpublished results and validate the hypotheses of the study, which serve to formulate policy measures to guarantee the application of CE during and after the pandemic. In addition, the motivation for carrying out this study is due to its contribution to the current state of the literature and to provide empirical evidence for policymakers' correct definition of policy aimed at achieving environmental sustainability.

Consequently, the contributions of the study are varied, which are indicated below. First, primary information sources are used from CE firms, which allows for obtaining direct information to assess the analysis situation better. In addition, it allows the investigation to be designed effectively, considering all the aspects to be examined. Second, the study uses stylized methods to test the hypotheses raised and maintain their scientific rigor. The applied methodology, CB-SEM, allows for obtaining efficient and unbiased statistics, which support the study's conclusions. Third, no studies have examined the relationship studied in the country of analysis, which makes it an unpublished investigation. Additionally, the study contributes to understanding how COVID-19 has influenced the decision to implement CE in a developing country. Fourth, there is little empirical evidence on the implementation of CE in developing countries [20]. Therefore, this research contributes to the literature by providing evidence on how to adopt CE in developing countries. These aspects show signs of concrete actions to achieve the environmental sustainability that the planet requires. After the introduction, the research structure is as follows: Section 2

describes the recent literature and the definition of hypotheses; Section 3 describes the data and methodological approach; Section 4 discusses the data and results; Consequently, Section 5 discusses the results obtained; furthermore, Section 6 contains the conclusions and policy implications.

## 2. Literature Review

### 2.1. Industry 4.0 and Circular Economy Practices

The CE focuses on the use of resources in the 10Rs to generate economic and environmental benefits [5]; however, some obstacles are related to the lack of advanced technologies to apply this circular flow [21], as well as the uncertainty of the economic benefit to be obtained from the investment made [22,23]; however, the emerging technologies of I4.0 are dynamizing the unlocking of CE and the collaborative economy, respectively [4].

Industry 4.0, known for the adoption of technology to maintain the efficiency of the firm [14], uses systems such as "Adoption of smart factory components", "Integration of digital and physical systems", "Environmental Product Design and life cycle analysis", and Adoption of advanced machine learning algorithms "in order to improve the performance of resources in the production process" [24]. These improvements include saving time in the processing of a product, reducing the cost of the final product, integrating the value chain, the resilience of the production process, making processes more flexible, and focusing on the efficiency of useful resources [25]. Consequently, the application of I4.0 technologies has gained great space in industrial transformation due to their great benefits for the firm and the environment [26]. Hence, I4.0 technologies contribute to applying the CE 10Rs; for example, they promote the reuse of discarded products, restore old products and update them, or design remanufactured products [27].

Indeed, blockchain technology contributes to the firm's CE since its application reduces transaction costs, improves performance and communication in the supply chain, generates a waste reduction, and, consequently, reduces carbon footprint [28]. In Indonesia, Ref. [29] states that the collection, transport, and processing of commercial waste presents CE processes efficiently achieved through the internet of things (IoT). Moreover, in South Africa, Ref. [30] mentions that the analysis of big data, information technologies and human capital are essential factors of I4.0, which are positively related to sustainable production constitutes a determinant for the CE.

Besides, Ref. [31], through structural equation modeling (SEM) in South Africa, find that the use of cloud computing, IoT, smart objects, GPS, radio-frequency identifications devices (RFID), among others, contribute to the application of CE, such as the design of products for reuse and recycling and reduction of solid waste and wastewater management costs, among others. In the same direction, Ref. [16] mention that the various tools of I4.0 allow improving the configuration of the production process, obtaining; as a result, the decrease in the total cost of production and the energy consumption of the machinery. On the other hand, Ref. [32] carried out a study in Estonia through semi-structured interviews; their results reveal that additive manufacturing and Big Data and Advance Analysis in all CE practices.

In addition, from a global perspective, the authors of Ref. [33] carry out an analysis with the heads of CE projects in Europe, America, Asia, and Africa. Their results show that I4.0 directly influences the EC, especially reducing the consumption of materials, energy, waste generation, and emissions. They also indicate that additive manufacturing and robotics are the technologies with the greatest impact on the EC. Finally, Ref. [34], through an in-depth review of the literature, indicate that I4.0 are the drivers to implement CE processes and achieve environmental sustainability. Therefore, we developed the following hypothesis:

**Hypothesis 1 (H1).** *Industry 4.0 technologies contribute to circular economy practices.*

### 2.2. COVID-19 and Circular Economy Practices

Before the outbreak of COVID-19, the CE was considered to be a strong economic approach for the future, but the crisis modified this approach, showing initiatives to incubate CE ideas to challenge the economy's linear model [6]. The COVID-19 outbreak has shown that the global economic model has been developing in a linear way in which the obtaining of economic benefits has prevailed without taking energy consumption into account, which is why the pandemic has become a trigger to bet towards a low carbon economy based on the CE [22,35]. Ref. [31] mentions that the COVID-19 outbreak has demonstrated the importance of implementing CE in production processes to mitigate the economic losses generated by a pandemic. Similarly, in Ref. [36], it is affirmed that the COVID-19 outbreak generated economic losses in companies. However, COVID-19 was an opportunity to apply CE, which allowed them to improve the product delivery service using fewer resources.

Ref. [2] mentions that the COVID-19 outbreak has elucidated the existing deficiencies in the supply chains' consumption and production processes. The current pandemic becomes an opportunity to improve them through CE, enabled by blockchain technology. In this way, sustainability becomes a competitive advantage for firms that seek to remain in the market in the current crisis since the COVID-19 outbreak has prompted firms to design new goods and services through the efficient use of resources and adequate waste management [9,10]. In the same sense, Ref. [32] indicates that due to the lockdown measures implemented during the COVID-19 outbreak, the energy sector has experienced a significant contraction, except for renewable energy consumption, which increased by 3%; specifically, this has been seen in renewable bioenergy, which is has a circular focus with zero waste.

In contrast, lockdown and mobility restriction measures have led to disruption in the shipping recycling supply chain in Bangladesh, India, and Pakistan, due to a lack of workforce to recycle ship components at the end of their useful life, representing a loss of around USD 20 billion [33]. Ref. [34] establishes that the pandemic has increased the need to take care of oneself with the use of single-use products (plastic water bottles, gloves, face shields, among others), which puts environmental sustainability at risk due to the large amount of waste that is generated, especially in low-income economies where there are linear economies.

Conversely, the pandemic has increased the demand for single-use plastic personal protective equipment to prevent the spread of COVID-19; however, they have become a polluting factor due to indiscriminate waste and added to the absence of CE for waste reduction, recycling, and recovery [36,37]. Similarly, [38] mention that the demand for plastic personal protective equipment, food, and groceries packaged in plastic has increased due to the COVID-19 outbreak; thus, CE, through recycling, constitutes one of the solutions to counteract the effect of plastic on the environment. Consequently, considering previous studies, Hypothesis 2 is defined:

**Hypothesis 2 (H2).** *The COVID-19 outbreak does not drive circular economy practices.*

### 2.3. Institutional Regulation and Circular Economy Practices

Concerns about environmental degradation have led governments to the establishment of policies that force firms to adopt clean, sustainable, and circular production systems [39].

Therefore, the government's regulatory role is an essential element when evaluating the adoption of CE motivated by institutional change [40]. However, there is a diversity of findings and criteria on institutional regulation's role in adopting the CE.

For this reason, Ref. [41], in a study of 60 countries, finds that the definition of government policies for the adoption of CE is making notable progress; however, the policies' efficiency is not according to expectations. For example, in a study conducted in Ref. [42] found that entrepreneurs do not trust the government to lead environmental

development towards an CE; however, they believe that business incubators would help in this process. In Tanzania, Ref. [43] mentions that environmental laws and regulations are not aimed at promoting CE; on the contrary, they exacerbate resource shortages, and moreover, the private sector and NGOs have contributed initiatives to launch the CE.

Likewise, through SEM processes information collected from Brazilian industrial firms, in Ref. [17] the main findings affirm that the adoption of CE depend to a great extent on the decisions of owners/shareholders, the institutional regulatory pressure of the government is incipient in determining the practices of CE, and ISO 14001 and ISO 9001 certification contribute to better CE adoption. Moreover, Ref. [13] identifies that institutional regulation constitutes a barrier for adopting a CE, given that the countries' institutional infrastructure favors the linear economy. Besides, international supply chains are highly coordinated, so that each country's heterogeneous policies become barriers to the adoption of CE practices [13].

Conversely, through SEM in the European Union (EU) Ref. [11] affirms that the institutional entrepreneurial role is key to adopting the CE; however, this varies according to the legislation of each EU country. Similarly, Ref. [44] found that the re-manufacturing industry in China has developed significantly, which has been achieved satisfactorily due to institutional and legal reforms and government policies implemented in recent years, and have contributed to green economic development, enacting a CE. Ref. [45] mentions that the reducing, reusing, recycling and repurposing processes are not borne of an initiative of the firm to achieve a CE, but as a product of institutional regulations; moreover, they assert that start-ups motivate the definition and modification of institutional regulations and legal regulations for the adoption of CE. Ref. [46] examines the role of institutional quality in adopting CE in EU countries using a structural equation model. Their results find that institutional promotion is a driving factor for the adoption of CE. Likewise, Ref. [47] examines the implementation of the Chinese CE through institutional policies through a systematic review of the literature. Their results show that the EC was notably boosted when China entered the Year Plan period (2016–2020). Finally, Ref. [48] examines how EU policies contribute to the adoption of CE. Their findings show that this type of institutional regulation has contributed notably to the CE, such as reducing waste. According to the empirical evidence described, Hypothesis 3 is defined as follows:

**Hypothesis 3 (H3).** *Institutional regulation contributes to the adoption of a circular economy practices.*

Consequently, Figure 1 synthesizes the hypotheses defined between constructs, which will be examined in the present investigation.

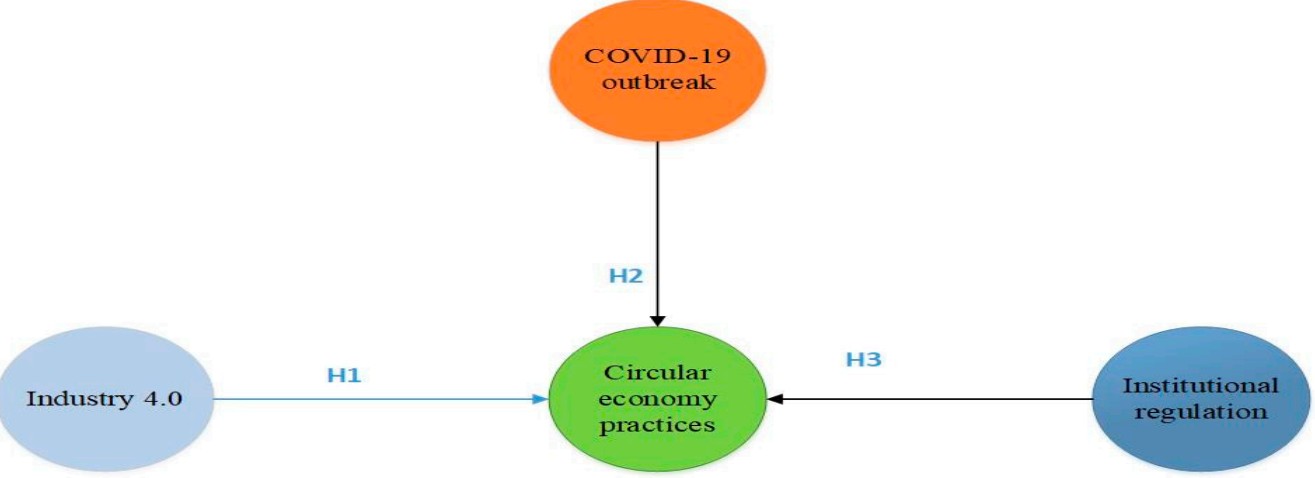

**Figure 1.** Proposed model.

## 3. Data and Methodological Approach

Through a questionnaire, information was collected from 214 manufacturing firms from different cantons of the four provinces contributing the highest percentage to the country's national production. The firms are from the private sector, considered as big firms. According to official statistics published by Ref. [49], 27.1% of the Gross Value Added (GVA) of manufacturing comes from the province of Guayas, 24.2% from Pichincha, and to a lesser extent it attributes Manabí with 5.5% and Azuay with 4.9%. Table 1 shows the distribution of the information collected.

**Table 1.** Demographical data collection.

| Provinces | Numbers | Cities | Numbers |
| --- | --- | --- | --- |
| Guayas | 88 | Guayaquil | 18 |
| | | Durán | 15 |
| | | Yaguachi | 10 |
| | | Coronel Marcelino Maridueña | 08 |
| | | Milagro | 11 |
| | | Sanborondón | 05 |
| | | Daule | 07 |
| | | Naranajal | 06 |
| | | Nobol | 03 |
| | | Santa Lucía | 05 |
| Pichincha | 45 | Quito | 20 |
| | | Rumiñahui | 10 |
| | | Mejía | 06 |
| | | Cayambe | 07 |
| | | Pedro Moncayo | 02 |
| Manabí | 36 | Montecristi | 25 |
| | | Manta | 09 |
| | | Jaramijó | 02 |
| Azuay | 30 | Cuenca | 12 |
| | | Gualaceo | 08 |
| | | Chordeleg | 06 |
| | | Sigsig | 04 |
| Total | 214 | Total | 214 |

A questionnaire was designed with several questions according to the examined constructs (see Appendix A) to gather the information from firms. The questionnaire is based on the Likert Scale (1 = strongly disagree, and 5 = strongly agree). The questionnaire was then applied online through Google forms. Consequently, the questionnaire was sent to the firms to be filled in. The information was collected from August to November 2020. After obtaining the information, the covariance-based structural equation modeling (CB-SEM) econometric approach was used in order to examine the relationship between constructs according to the information provided by their factors; likewise, it allows us to capture the effect of latent interaction and moderation [50]. One of the advantages of the CB-SEM approach is its particularity in investigating various relationships between interrelated constructs simultaneously [51]. Another advantage is that it improves the robustness of the estimators, optimizes the results for the interpretation of the interaction effects and efficiently controls the measurement error [52]. Conversely, the proposed model meets the minimum criteria for applying CB-SEM. CB-SEM can be used with samples greater than 200 observations [53,54], and likewise, CB-SEM offers the availability to use several constructs and with any number of constructed latent interaction indicators [55].

Similarly, to determine the suitability of the constructs to be used in the research, reliability and validity tests are performed to obtain robust and unbiased results using exploratory factor analysis (EFA) and confirmatory factor analysis (CFA) [56].

## 4. Analysis of Data and Results

After collecting the information of firms using the online questionnaire, the reliability and validity of the data must be confirmed. The questionnaire is designed with several questions (items) that represent each construct. The "items" column of Table 2 indicates the number of questions that each construct contains, which were sent to the firms for their answer. Therefore, the study uses the exploratory factor analysis (EFA) and confirmatory factor analysis (CFA). First, the EFA is applied to examine the unidimensionality of the constructs and, consequently, define the factors responsible for representing a thing in common, in other words, a set of diverse factors that explain a single concept [57]. Thus, Table 2 shows the results of factor loadings that are higher than 0.6, which supports the unidimensionality of the constructs of circular economy (CE) practices, industry 4.0 (I4.0) technologies, coronavirus disease 2019 (COVID-19) outbreak and institutional regulations for environment (IRE). Additionally, Cronbach's Alpha values are between 0.81 and 0.92, which are higher than 0.7 and provide an argument favoring internal consistency between the measurement elements within each construct.

**Table 2.** Instrument reliability and validity.

| Constructs | Symbol | Items | Factor Loading Ranges | Cronbach's Alpha | AVE | CR |
|---|---|---|---|---|---|---|
| Circular economy practices | CE | 12 | 0.741–0.892 | 0.91 | 0.637 | 0.864 |
| Industry 4.0 technologies | I4.0 | 11 | 0.759–0.934 | 0.92 | 0.696 | 0.905 |
| Coronavirus disease 2019 | COVID-19 | 5 | 0.68–0.873 | 0.87 | 0.671 | 0.833 |
| Institutional regulations for environment | IRE | 6 | 0.582–0.816 | 0.81 | 0.568 | 0.739 |

Subsequently, the CFA examines the factors included in the dimensions of the study; first, Composite Reliability (CR) is a more efficient measure of internal consistency; the values are higher than 0.7 and vary between 0.739 and 0.905 [58]. In addition, the average variance extracted (AVE) values are more significant than 0.5, which guarantees that the variables meet the convergent criteria, respectively. Consequently, the Fornell and Larcker criterion confirms the discriminant validity, as reported in Table 3. Then, results of the CFA that are displayed in Table 4.

**Table 3.** Discriminant validity analysis.

| Constructs | COVID-19 | CE | IRE | I4.0 | Mean | S.D. |
|---|---|---|---|---|---|---|
| COVID-19 | 0.822 | | | | 3.52 | 0.621 |
| CE | 0.515 | 0.798 | | | 2.98 | 0.495 |
| IRE | 0.379 | 0.261 | 0.753 | | 2.38 | 0.337 |
| I4.0 | 0.497 | 0.507 | 0.479 | 0.834 | 4.55 | 0.754 |

**Table 4.** Model fitness results.

| Fit induces | NNFI | NFI | CFI | GFI | AGFI | TLI | $\chi^2/df$ | RMSEA | SRMR |
|---|---|---|---|---|---|---|---|---|---|
| Criteria | ≥0.90 | ≥0.90 | ≥0.90 | ≥0.90 | ≥0.90 | ≥0.90 | ≤3 | ≤0.08 | ≤0.08 |
| Measurement model | 0.913 | 0.910 | 0.911 | 0.907 | 0.929 | 0.938 | 1.892 | 0.037 | 0.027 |
| Structural model | 0.922 | 0.922 | 0.934 | 0.912 | 0.932 | 0.941 | 1.282 | 0.038 | 0.026 |

Table 4 shows model fitness, whose values of the nine indicators show an adequate specification of the measurements of the variables as well as the model [59]. The indicators used are the non-normed fit Index (NNFI), normative fit index (NFI), comparative fit index (CFI), the goodness of fit index (GFI), adjusted goodness of fit index (AGFI), Tucker–Lewis index (TLI), Chi-square to the degree of freedom ($\chi^2/df$), root-mean-square error of approximation (RMSEA) and standardized root mean squared residual (SRMR).

The values of NNFI, NFI, CFI, GFI, AGFI, and TLI are greater than the limit established for each indicator following [60]. Besides, the value of χ2/df (1.892) is less than three and that of RMSEA (0.037) less than 0.08, which agrees with the criteria established by [61]. Additionally, the approximate fit, which measures the difference between the observed correlation matrix and the implicit correlation matrix of the model, is determined by SRMR, whose value (0.027) is less than 0.08 [62]. Based on what was previously analyzed, it is confirmed that the proposed model has a great fit with the data; therefore, the next step is to analyze the results of the hypotheses.

In this scenario, Figure 2 and Table 5 present the standardized regression estimates of variables in the structural model and the significance of path weights. The regression carried out was of the ordinal type, since each question has a response option from 1 to 5, according to the Likert Scale. Therefore, ordinal regression allows for the processing of responses of polytomous variables. I4.0 maintains a positive and significant relationship with CE ($\alpha = 0.716$, $p = 0.000$) as expected. Then, the effect of the COVID-19 outbreak is examined, in which it is identified that there is no strong or statistically significant association with CE ($\alpha = 0.871$, $p = 0.541$). This result is novel and was shown to be unknown due to the few precedents associated with the COVID-19 outbreak in the industry. Finally, the relationship between IRE and CE were examined, in which evidence is found in favor of the fulfilment of the hypothesis, IRE has a positive and statistically significant effect on CE ($\alpha = 0.834$, $p < 0.01$), whose results are found according to expectations. The results obtained provide important information that supports the fulfilment of the study hypotheses H1–H3.

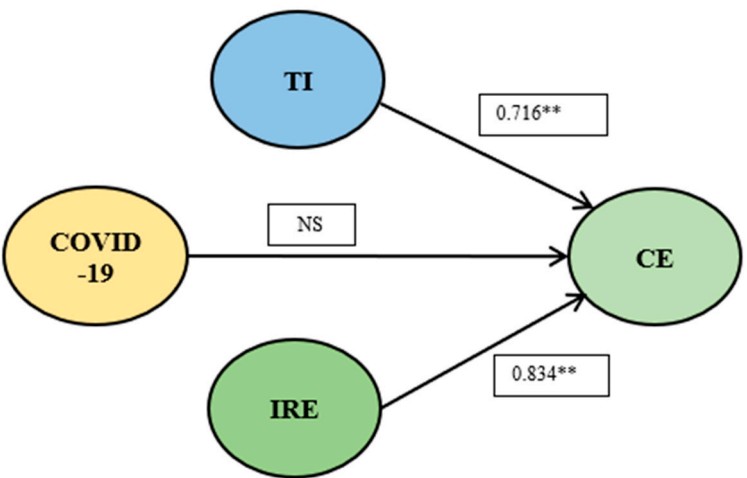

**Figure 2.** The results of the structural model. Note: ** indicates the significance.

**Table 5.** Standardized parameter estimates for structural model.

| Hypothesis | Paths | Standardized Estimate | *p*-Value | Results |
|------------|-------|----------------------|-----------|---------|
| H1 | I4.0 → CE | 0.716 ** | 0.000 | Supported |
| H2 | COVID-19 → CE | 0.871 | 0.541 | Not supported |
| H3 | IR → CE | 0.834 ** | 0.006 | Supported |

Note: ** indicates significance at 1% and 5% respectively.

## 5. Discussion of Results

The COVID-19 outbreak has directly impacted various economic activities worldwide due to the lockdown and mobility restriction measures that have been defined to mitigate the spread of the pandemic. Likewise, the it has generated a favorable scenario to promote the CE as an alternative to improve the efficiency of using resources, which brings with it a cleaner and more environmentally-friendly approach to production [6]. Consequently, this research examines the relationship between the technologies of industry 4.0, the COVID-

19 outbreak, institutional regulations for environment and circular economy practices in manufacturing firms in Ecuador. This fact constitutes a valuable contribution to the country of study due to the few studies that examine the determinants of CE.

The results shown in the previous section show the direct effects of the explanatory variables of the study on CE, which were described in Hypotheses 1–3. In the first place, I4.0 has a direct effect on the adoption of CE in manufacturing industries, from which it should be understood that those firms that have modern information systems, with state-of-the-art technology in their production processes, intelligent processing of data, among others, are easier to adopt and apply CE. This fact could be explained by the fact that a production process with a high component of I4.0 generates information in each phase for the adequate decision-making of the firm on the efficient use of resources, such as reducing waste, saving electricity and water, among others. Moreover, the reuse or re-manufacturing of inputs and products is more efficient with modern technological processes of I4.0. These findings coincide with the study by the authors Ref. [16], who affirm that the adoption of intelligent technologies, such as those of I4.0, contributes to increasing the efficiency of resources, reducing waste and saving energy in machinery. Similar to our findings, Ref. [26] affirms that the use of I4.0, such as IoT, GPS, or RFID drives the design of products for reuse or recycling, as well as improving waste management.

Next, the COVID-19 outbreak shows a positive sign but not statistically significant. In other words, the pandemic was able to corroborate that a linear system of the economy does not contribute to environmental sustainability due to the excessive amount of waste and inefficiency of resources used in the production of goods and services. Thus, several firms have been inclined to carry out CEs before the onset of the COVID-19 outbreak. However, the research does not find evidence in favor of COVID-19 having caused the implementation of CE in Ecuadorian firms, which could be explained for two main reasons. First, firms that adopt CE do so in an organized and planned way, considering the firm's performance, not because of a risk situation due to mitigation measures for the spread of COVID-19. Thus, for example, manufacturing firms did not have the necessary response capacity to adapt to this situation, and on the contrary, their planning towards a circular economy becomes uncertain [63]. Second, COVID-19 generated a situation of uncertainty in the firm performance, which causes firm managers to have greater doubts when it comes to adapting the firm to apply CE in its production processes since it is required of a significant investment which becomes riskier in the middle of a pandemic as suggested by [64]. Thus, the results obtained in this study are contrary to those found by the author of Ref. [9], who mentions that the COVID-19 outbreak has been the driving force for firms to design new products and improve waste management by adopting CE.

Conversely, IRE presents a positive and significant effect on CE. This result must be associated with the important role that the government plays, through its environmental regulation policies, in order for the production processes of manufacturing firms to be clean and not to degrade the environment. However, government intervention may not achieve the environmental solutions that an economy requires [17]. These types of regulations, such as ISO 4001 or ISO 9001, aim to promote the reduction of the environmental impact resulting from industrial activity, for which it is recommended to achieve environmental sustainability through the adoption of CE. Similar findings to those found by Ref. [65] in China indicate that the Chinese re-manufacturing industry has achieved sustainable development with CE, thanks to the institutional and legal reforms of the country in recent years. On the contrary, Refs. [13,66] state that IRE becomes a public barrier to adopt the CE because the manufacturing supply chains are internationally coordinated, and environmental policies are different for each country.

## 6. Concluding Remarks and Policy Implications

CE is a commitment to sustainable development to change the linear model of the economy, causing the deterioration of the environment. However, the adoption of the CE is an arduous task that requires all economic agents' effort, especially government

leadership, to apply measures that force firms to adopt cleaner production systems. The adoption of CE has taken on great importance in the current pandemic that the world is going through since it is necessary to make the most of productive resources to deplete the supply chain of the manufacturing sector. In this sense, this study has found essential and unpublished findings on this particular topic by collecting primary information from firms in the Ecuadorian manufacturing sector, which was subsequently processed with CB-SEM. The results find evidence favoring the existence of direct effects of I4.0 and IRE on the CE. However, there is no evidence favoring COVID-19 being a determinant of CE adoption in manufacturing firms in the country. Based on the results obtained from the study, the following policy implications emerge:

- The Central Government should lead the implementation of I4.0 in manufacturing firms with a legal project that is progressively achieved, especially in industries where they have a mostly marked linear economy approach.
- The Central Government must provide facilities for firms to acquire I4.0 technological equipment, such as 0% tariffs for importing this type of equipment. Similarly, the State must guarantee loans with low-interest rates so that firms can finance such technological acquisitions.
- The manufacturing supply chain must be promoted by adopting I4.0 so that the CE is not only an isolated practice of specific firms, but on the contrary, it is a green practice of all the agents that participate in the chain of supply.
- Tax incentives or subsidies must be generated to those firms in which their production has been achieved through CE and generate public-private alliances to open local and international markets to commercialize these products. In the same way, incentives must be generated in consumers to direct their choice of consumption to the goods that come from a production process through CE.
- The government must give a more leading role and increase the powers of the institutions of environmental regulation and progressively increase the penalties for firms that continuously degrade the environment, due to the obsolete production process.
- Likewise, there should be a greater presence of these institutions to improve the efficiency of regulation and learn about each industry's particularities to improve the transition to manufacturing firms with CE.
- The COVID-19 outbreak does not represent a determining factor for the adoption of CE; however, firms should have management plans to face that presents uncertainty and risk to be able to adapt the production process in these scenarios and be ready to apply the 10R of CE.

Like most studies, one of the limitations is that the level of detail of the information is limited. However, despite the study's limitations, the contribution is unprecedented and constitutes one of the significant pioneering studies on the subject in the country and the region.

**Author Contributions:** Conceptualization, S.A.R.K., P.P. and M.T.; methodology, M.S.A.-A., Z.Y. and G.T.; software, Z.Y. and P.P.; validation, S.A.R.K., G.T. and M.S.A.-A.; formal analysis, P.P. and Z.Y.; investigation, S.A.R.K. and M.T.; resources, P.P. and M.S.A.-A.; data curation, P.P. and Z.Y.; writing—original draft preparation, S.A.R.K., M.T. and P.P.; writing—review and editing, G.T., M.S.A.-A. and Z.Y.; visualization, P.P. and Z.Y.; supervision, S.A.R.K., G.T. and M.T.; project administration, S.A.R.K., M.T.; funding acquisition, S.A.R.K. and G.T. All authors have read and agreed to the published version of the manuscript.

**Funding:** This research is supported by the Beijing Key Laboratory of Urban Spatial Information Engineering (No. 20210218) and the APC funded by Prince Sultan University.

**Acknowledgments:** This research is supported by the Beijing Key Laboratory of Urban Spatial Information Engineering (No. 20210218). Furthermore, all authors of this article would like to thank the Prince Sultan University for its financial and academic support to publish this paper in *"Sustainability"*.

**Conflicts of Interest:** The authors declare no conflict of interest.

## Appendix A. Questionnaire

*Constructs and Questions*

**Industry 4.0**

Advanced technology contributes to inefficient decision-making.
Artificial intelligence applications help to reduce polluting activities.
Advanced technologies help in the processing of large amounts of data.
Technology allows obtaining reliable information on the production process.
The adoption of smart technology improves the use of resources.
The use of technology makes it possible to improve circular economy practices.
The use of technology improves the efficiency of the firm.
Advanced technology allows to redistribute product delivery and reduce waste.
Technology contributes to the green design of products.
Technology allows for improving the firm's response to adverse natural events.
Technology allows flexibility and improvement of the operational processes of the firm.

**Circular Economy Practices**

Design of Products for Reduced Consumption of Material/Energy.
Design of Products for Reuse, Recycle, Recovery of Material, or Components Parts.
Design of Products to Avoid or Reduce Use of Hazardous Products & Their Manufacturing Process.
Ensure suppliers meet their environmental objectives.
Requires suppliers to have certified EMS like ISO 14001.
Ensure purchased materials contain green attributes.
Requires suppliers to develop and maintain an EMS.
Decreasing toxic and hazardous chemicals in manufacturing processes.
Reducing fossil fuel energy consumption.
Using green materials in manufacturing.
The firm's production process prioritizes the consumption of raw materials and energy.
The firm's initiative improves the energy efficiency of production equipment.

**Institutional regulations for the environment**

Implementation of carbon-taxation.
Heavy penalties and fines due to violation of environmental policies.
Safety training to the employees.
Environmental awareness training for the labor-force.
Zero and/or low-interest-rate loans for the circular economy projects.
Tax exemption policies for green projects.

**COVID-19 Pandemic**

Lockdown creates pressure on firms to adopt eco-friendly practices.
During the pandemic, online delivery of products improves long-lasting sustainability.
Due to the COVID-19 pandemic, firms efficiently consume energy and water to increase sustainability.
Lockdown policies are a solution to increased environmental performance.
During the pandemic, firms adopting teleworking patterns in the context of sustainability.

*Note: This questionnaire is used to collect the data from the manufacturing firms. The questionnaire is based on the Likert Scale (1 = strongly disagree, and 5 = strongly agree)*

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
