# Peer review of "Digital Technologies, Circular Economy Practices and Environmental Policies in the Era of COVID-19"

_sustainability, doi:10.3390/su132212790_

Round 1
Reviewer 1 Report
The paper is interesting and make an evaluation of the situation in Ecuador about Digital Technologies (not very clear in the paper), Circular Economy Practices and Environmental Policies in the Pandemic COVID-19 period. We are living yet this tragic period of time, unfortunately...
There are some few mistakes in the paper: we need to correct the application of the acronym CE (many times you are using EC). The Figure 1 is shown but never referenced in the text. The same to Figure 2. The Figure need to by identified the source (is it from the author or extracted from other research?).
The sample consists of 214 companies from Ecuador. Ok, but the author does not describe from which sector, if they are SME or big companies, private or public, etc.
The statistical analysis looks well done and the Hypothesis are well presented (the theoretical framework is done before the formulation of the hypothesis).
In the abstract the authors said:
The results suggest that Industry 4.0 technologies and environmental regulation policies drive circular economy practices during the pandemic. The study finds no evidence favoring COVID-19 being a determining factor in the adoption of the circular economy. The results provide a policy framework for the adoption of a circular economy. It is validated in the work, ok.
There is a lack about the last legislation in European Union about wastes. It is not well presented and outdated. For example, in Portugal there is new regulation about wastes and national policies.
This paper needs some improvements, but is a good contribution for this specific field.
Author Response
Dear, Thanks for your valuable comments and suggestions.

Reviewer 2 Report
The theme is very current and interesting. Many thanks for this interessting journal article. You did a good methodical work with interessting outcomes. The article is well stuctured and written in good English. It is easy to follow your research.
You should take the following issues into account:
- Introduce Figure 1 and explain it with a brief summary description of what has already been described previously.
- Figure 2 is also not introduced. I suggest introducing and describing Figure 2 as well.
- The presentation of the hypotheses H1, H2 and H3 is unclear. Please introduce and present them more clearly.
Author Response
Really appreciated your value-added comments.

Reviewer 3 Report
Dear authors,
thank you for handing in your manuscript “Digital Technologies, Circular Economy Practices and Environmental Policies - In the Era of COVID-19“ and for giving me the chance to review it.
The topic of sustainability and of cirular economies is of present and future importance on a global level. The authors conduct a survey among Ecuadorian companies to answer three hypothesis.
However, I want to highlight aspects to further strengthen the manuscript:
Introduction:
- The introduction is to vague & does not lead to the core of the topic.
Literature review:
- In my view, the literature review does not properly support the hypothesis H1. 3 resources (1x Indonesia, 2x South Africa) are very little as support.
- For hypothesis H2, there seems to be no support at all. As the authors describe it, „has become a trigger to bet towards a low carbon economy.“ After that, it is explained that circular economy came to a hold in Pakistan & Bangladesh, as well as in other parts of the world regarding face protection. I do not see, how the authors developed hypothesis H2.
- Hypothesis 3: „Also, [13] identify that institutional regulation constitutes a barrier for adopting a CE, given that the countries' institutional infrastructure favours the linear economy“ – Is this not the same as given in the authors‘ hypothesis 3 while just looking on the other side of the scale?
Data & Methodology:
- Applying SEM requires to fulfil a number of criteria. Are all these criteria met? E.g. small sample size? Observation? Number of input variables considered?
- Information: „After collecting the information“ – which information was collected & how was it gathered?
Analysis:
- Analysis: What was analysed? What did the authors ask the firms for?
- Robustness of the research: Is the chosen sample in any way representative for Ecuador?
- Table 1: Please check the format.
- Table 2: What are the „Items“?
- Table 3: Which regression?
Discussion chapter:
- How can the authors claim that„… several firms have been inclined to carry out CE.“ If the say further that „…, the study findings do not find evidence favouring this claim“?
Overall the whole article is really vague. It does not explain what was done, how it was done and how the authors gathered the information & analysis. Further, it is, until now, poorly supported.
Formal aspects:
- Minor English language & grammar check is required. Some sentences seem to be not complete.
- Figures: No source given. Are these developed by the authors of the article?
- Please define the abbreviation „EC“. Or is it the same as „EC“?
I with the authors all the best
Best Regards
Author Response
Dear, Bottom from the heart, thank you for your comments and suggestions. We have done accordingly. Thank you again.

Reviewer 4 Report
Dear authors,
I have read your paper twice, and I can say I like it.
Well formulated, analyzed. I only suggest to improve your literature (little things) :
a) 10.7232/iems.2020.19.3.551;
b) 10.3390/app10030755;
for the rest it's ok.
Author Response
Thank you very much for your kind comments and suggestions. Kind Regards

Round 2
Reviewer 3 Report
Dear authors,
thank you for handing in your revised manuscript “Digital Technologies, Circular Economy Practices and Environmental Policies - In the Era of COVID-19“ and for giving me the chance to review it once again.
Once reading through it, I have to admit the script has been significantly improved. Anyway, there are still a few topic I would like to mention:
Figure 1: Did the authors obtain the licence from the initial authors that they are aloud to copy the figure? Otherwise, I assume a copyright issue.
Introduction:
- The introduction does not show what is the motivation for the study and does therefore not lead to the core of the topic. It should also tell why especially Ecuador is so interesting for such a study to be published on international level.
Literature review:
- Hypothesis 2 should be “consequently-developed“. Reading the paragraph over and over, I am not able to come the same conclusion as the authors. I would assume hypothesis 2 to be the opposite. Even more, as single-use plastic consumption increased. The results, eventually, show the same.
- Hypothesis 3: 3 sources is not much to support the hypothesis.
Analysis:
- Table 2: Do the „Items“ refer to the number of questions asked or to the number of companies answering? I think, this should be stated clearly in the text as additional information to the table.
- „In this scenario, Figure 3 and Table 5 present the standardized regression estimates“. The authors should add which regression method they have used & why it should be appropriate for the set of data obtained. E.g. logistic regression, numerical regression, ordinal regression,…?
Overall merit: This is a much better version. However, in some areas the authors should be more specific a) with focus to the topic, b) about the methods used.
Formalities:
- Minor English language & grammar check is required. Some sentences seem to be not complete.
- Figures: Please check whether there is a copyright issue with figure 1.
I with the authors all the best
Best Regards
Author Response
Dear Respect Reviewer,
Thanks for your kind suggestions, we have done accordingly and hope you will find a better manuscript.
Kind Regards
